# Mpox due to monkeypox virus clade Ia infection detected outside of Africa: a case study
Cian Dowling-Cullen [1,11] ✉, Laura Fahey[2,11], Michael Carr [2,3], Brian Keogan[2], Josephine Hebert[4], Siti Mardhiah Muhamad Fauzi[5], Jane Stapleton[6], Allison Deane[6], Elizabeth O'Donoghue[7], Colette Reilly[7], Christine Kelly [5,8], Eavan G. Muldoon[5], Margaret O'Sullivan[6], Anne Sheahan[6], Suzanne Cotter[7], Mary O'Meara[7], Eoghan De Barra[4], Éamonn O'Moore[1], Keith Ian Quintyne [1,9], Cillian De Gascun[2], Áine O'Toole [10], Daniel Hare [2,12] & Derval Igoe[1,12]

## Abstract

**Background** Monkeypox virus clade Ia has historically been restricted to central Africa and primarily associated with zoonotic transmission. In February 2025, a symptomatic adult with recent travel to the Democratic Republic of the Congo presented to a tertiary care facility in Ireland and was subsequently diagnosed with monkeypox virus clade Ia infection. We describe the clinical, laboratory, phylogenetic and epidemiologic features of this case.
**Methods** Diagnostic testing was performed on the 5th of February using real-time polymerase chain reaction. The patient was admitted to the National Isolation Unit, and comprehensive contact tracing with active surveillance was implemented. Whole-genome sequencing was performed to determine the viral lineage and conduct phylogenetic and mutation analyses.
**Results** The patient experiences a stable disease course and recovers without long-term complications. No secondary transmission is detected. Genomic analysis confirms infection with monkeypox virus clade Ia. The sequence clusters within the recently documented sh2024a outbreak in Kinshasa and displays APOBEC3-associated mutational signatures consistent with sustained human-to-human transmission.
**Conclusions** This case highlights changes in the global epidemiology of monkeypox virus clade Ia and underscores the need for enhanced surveillance and coordinated international public health responses. Early detection and monitoring by whole-genome sequencing are essential for mitigating the risks posed by possible expansion beyond traditionally endemic areas.

## Plain language summary

Mpox is an infectious disease caused by the monkeypox virus (MPXV). MPXV clade Ia is one type of MPXV. Until recently, MPXV clade Ia infections mainly occurred in parts of central Africa and human infections were mainly transmitted from animals. In February 2025, a case of MPXV clade Ia infection was identified in a person returning to Ireland from the Democratic Republic of the Congo. Here, we describe the clinical features and laboratory results of the case. This case report draws attention to recent changes in the transmission of this virus with spread to regions not previously affected. This highlights issues around the importance of monitoring MPXV clade Ia and other viruses, including through laboratory-based surveillance.

The first human infection with monkeypox virus (MPXV) clade Ia was reported in 1970 in the Democratic Republic of the Congo (DRC)[1]. Historically, most MPXV clade Ia infections have occurred through zoonotic transmission, with limited secondary human-to-human spread[2,3]. The majority of cases have been detected in the DRC, with fewer cases reported in other central African countries[2]. The exact details of the mpox animal reservoir are uncertain, but research suggests that rodents are likely involved[2,4,5].

The mean incubation period of MPXV clade Ia has been estimated at approximately 10 days[6]. A review of evidence up to 2020 estimated the case fatality rate (CFR) for MPXV clade Ia at approximately 10.6%[7]. However, more recent CFR estimates have been lower[2], and these estimates may be

[1]National Health Protection Office, Health Protection Surveillance Centre, Dublin, Ireland. [2]National Virus Reference Laboratory, University College Dublin, Dublin, Ireland. [3]International Collaboration Unit, International Institute for Zoonosis Control, Hokkaido University, Sapporo, Japan. [4]Beaumont Hospital, Dublin, Ireland. [5]National Isolation Unit, Mater Misericordiae University Hospital, Dublin, Ireland. [6]Department of Public Health HSE South West, Cork, Ireland. [7]Department of Public Health HSE Dublin and North East, Dublin, Ireland. [8]Centre for Experimental Pathogen Host Research, University College Dublin, Dublin, Ireland. [9]School of Population Health, Royal College of Surgeons in Ireland, Dublin, Ireland. [10]Institute of Ecology and Evolution, University of Edinburgh, Edinburgh, United Kingdom. [11]These authors contributed equally: Cian Dowling-Cullen, Laura Fahey. [12]These authors jointly supervised this work: Daniel Hare, Derval Igoe. ✉e-mail: cian.dowlingcullen@hse.ie

impacted by factors such as population demographics, healthcare access, and testing or reporting practices[8]. Regarding specific treatments, antivirals are under evaluation, but a recent clinical trial in the DRC did not demonstrate improvement in lesion resolution with tecovirimat[9]. A World Health Organization (WHO) clade Ia risk assessment reports cross-protection from previous orthopoxvirus infection or vaccination is likely, with moderate confidence in this assessment due to limitations in the available evidence[2].

Dramatic changes in mpox epidemiology have been observed in recent years. In 2022, a global outbreak caused by MPXV clade IIb prompted the WHO to declare a Public Health Emergency of International Concern (PHEIC)[10]. In August 2024, another PHEIC was declared, this time in response to the newly identified MPXV clade Ib and the associated surge in cases in the DRC, and spread to neighbouring countries[11]. Sporadic MPXV clade Ib infections have since been detected outside Africa, mainly associated with travel from central Africa[12].

Amid these clade IIb and clade Ib events, which have drawn considerable global attention, the epidemiology of MPXV clade Ia has also undergone significant recent changes. In particular, a shift toward increased human-to-human transmission has been demonstrated in Kinshasa in the DRC, raising regional and international public health concerns[13,14]. While the pathways for human-to-human transmission in this outbreak are uncertain, sexual contact may be involved[13,14]. The WHO recently assessed the overall global public health risk associated with MPXV clade Ia as moderate and noted that the outbreak in Kinshasa requires particular attention[13].

Genomic analysis has played an important role in the detection and evaluation of these emerging shifts in MPXV clade Ia epidemiology. One source of evidence supporting sustained human-to-human transmission of MPXV is the accumulation of mutational signatures consistent with apolipoprotein B mRNA editing enzyme, catalytic polypeptide-like 3 (APO-BEC3) activity. These are antiviral enzymes in the human innate immune system that respond to infection by inducing specific mutations (TC → TT on the leading and GA → AA on the lagging strands) of viral genomes. The observed accumulation of APOBEC3 mutations in the viral population can thus be employed to infer the duration and extent of human-to-human transmission[15].

In the context of these recent changes in MPXV clade Ia noted in the DRC, a case of MPXV clade Ia infection was notified in February 2025 in Ireland, in an adult male returning from the DRC. To the best of our knowledge, this represented the first confirmed case of MPXV clade Ia infection outside Africa to be reported internationally at the time. Here, we describe the clinical, laboratory, epidemiologic, and phylogenetic features of this case as well as the public health response, highlighting the growing global significance of MPXV clade Ia.

## Methods
### Case description and investigation
On the 4th of February 2025, a 39-year-old man presented to an Emergency Department (ED) in Dublin, Ireland, with a one-day history of a pustular rash and a six-day history of fever, sore throat and subauricular swelling. The rash had begun on the face and had spread after one day to the trunk and upper limbs. The patient had arrived in Ireland earlier that day, travelling via Türkiye after a four-week visit to the DRC, and proceeded directly from the airport to the ED (Fig. 1). While in the DRC, he had stayed in Kinshasa and in a nearby village, frequently travelling between these two locations. Potential exposures in the DRC included close contact with adults and children during volunteer work. Additionally, there was exposure to animals, including contact with ostriches and the hunting and consumption of antelope. Furthermore, the patient slept outdoors without shelter while in the village, raising the possibility of unwitnessed exposure to rodents or other animals. There were no reported sexual exposures. The patient was previously healthy and had no history of orthopoxvirus vaccination. At initial presentation to the ED, the patient reported no contact with other individuals who had similar symptoms. One adult contact in the DRC subsequently developed a fever, but no specific clinical or laboratory diagnosis was confirmed for that individual.

Upon examination, the patient was afebrile and exhibited evidence of a widespread rash on his face, trunk, and upper limbs. This rash was vesicular and pustular in nature, with crops of sub-centimetre, non-confluent and umbilicated lesions at varying stages of eruption (Fig. 2). There was also significant palpable unilateral subauricular lymphadenopathy and evidence of associated tonsilitis with overlying pus.

Given the travel history and clinical findings, the patient was isolated after triage and was reviewed urgently by the local infectious diseases service, who promptly notified public health authorities due to suspicion of mpox. Initial laboratory workup revealed elevated creatinine and gamma-glutamyltransferase (GGT) (Table 1). Additionally, the patient was found to have previously undiagnosed type 2 diabetes and hypertension. A swab of a skin lesion was sent to Ireland's National Virus Reference Laboratory (NVRL) for quantitative real-time polymerase chain reaction (qPCR) testing for orthopoxiruses, herpes simplex virus, and varicella zoster virus. The patient was admitted to hospital and commenced on intravenous (IV) ceftriaxone and dexamethasone for a suspected superimposed bacterial tonsilitis.

On the 5th of February 2025 (day seven of illness), qPCR results confirmed MPXV clade I infection, with strong suspicion for clade Ia due to a negative clade Ib-specific result. Based on these findings, the patient was transferred to the National Isolation Unit, a facility in a Dublin university teaching hospital that provides specialised care for patients with high-consequence infectious diseases, in line with public health advice at the time. Antimicrobials were de-escalated to oral co-amoxiclav at this point to complete a seven-day course for bacterial tonsilitis, and the IV dexamethasone was changed to oral prednisolone. By day eight of illness, the was still experiencing eruptions of new lesions, with sub-centimetre papular and vesicular lesions appearing on his right inner thigh, both knees, and the dorsum of his left foot. On day nine of illness, mild conjunctival injection was noted and some of the older lesions had crusted over. No new lesions developed after day nine of illness. Approximately 30–50 lesions were observed in total.

The patient remained clinically stable throughout hospital admission and pain management was optimised. On the 10th of February 2025 (day 12 of illness), whole-genome sequencing (WGS) results confirmed MPXV clade Ia infection. Repeat laboratory testing on day 12 of illness demonstrated resolved renal impairment and improved GGT. On day 14 of illness, given his stable clinical condition, and in consultation with public health authorities, the patient was transferred to a self-care isolation facility outside the acute hospital system for the remainder of the infectious period. De-isolation and discharge planning followed Irish guidance at the time[16], which was adapted from the UK Health Security Agency (UKHSA) guidance[17]. Discharge was approved at a multidisciplinary team meeting once the patient met clinical criteria (clinically safe for discharge), laboratory criteria (negative PCR results from urine and throat swab), and lesion criteria (no new lesions for 48 h, no mucous membrane lesions, all lesions to have scabbed over, the scabs dropped off, and a fresh layer of skin to have formed under all previous lesions). The patient was discharged home on day 20 of illness, on the 18th of February 2025.

### Public health response
A multidisciplinary mpox National Incident Management Team (NIMT) had been stood up by Ireland's National Health Protection Office since the WHO PHEIC declaration in August 2024, to prepare for potential imported cases. This NIMT coordinated the public health response to this case, which included case isolation, contact tracing, post-exposure prophylaxis, and national and international communications.

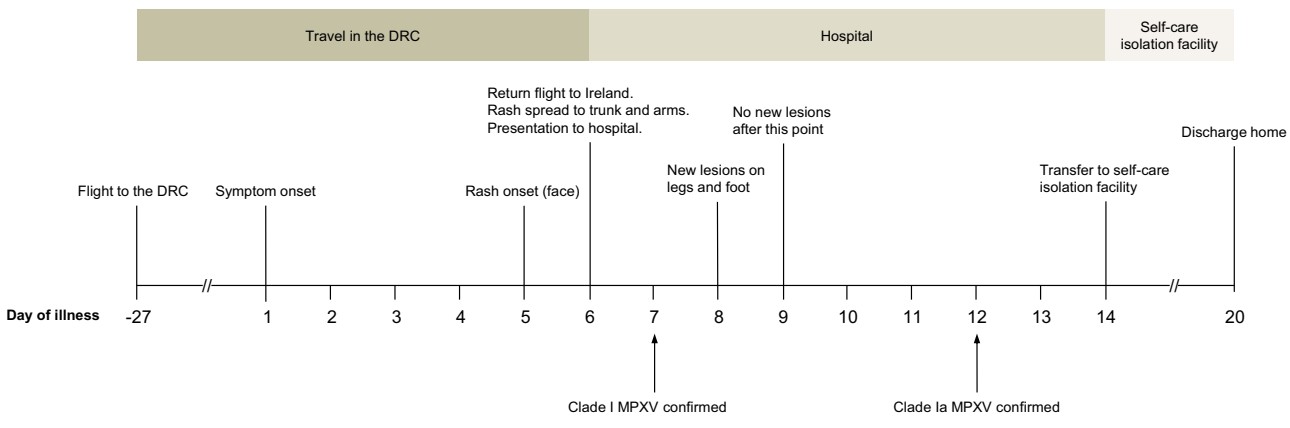

**Fig. 1 | Timeline.** Timeline of the main events relating to this case.

**Fig. 2 | Mpox lesions.** Photographs of mpox lesions on the right lower limb taken on day 6 of illness.

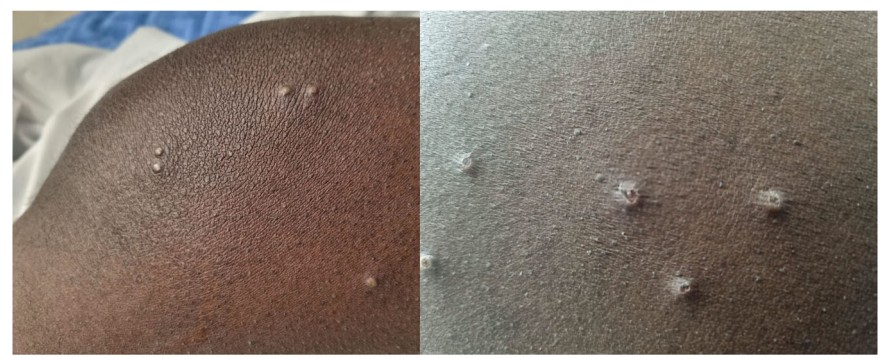

## Table 1 | Clinical laboratory results at presentation to hospital on day 6 of illness

| Measure | Reference range* | Result |
|---|---|---|
| White cell count (per µL) | 4000–11,000 | 12,070 |
| Red cell count (per µL) | 4,000,000–6,500,000 | 4,980,000 |
| Absolute neutrophil count (per µL) | 2000–7500 | 5870 |
| Absolute lymphocyte count (per µL) | 1000–4000 | 4890 |
| Platelet count (per µL) | 140,000–400,000 | 207,000 |
| Haemoglobin (g/dL) | 13.0–17.5 | 12.8 |
| Haematocrit (%) | 37.0–54.0 | 38.1 |
| Sodium (mmol/L) | 133–146 | 145 |
| Potassium (mmol/L) | 3.5–5.3 | 3.3 |
| Chloride (mmol/L) | 95–108 | 103 |
| Creatinine (mg/dL) | 0.67–1.18 | 1.74 |
| Blood urea nitrogen (mg/dL) | 7.8–22.7 | 13.16 |
| Total protein (g/dL) | 6.0–8.0 | 7.5 |
| Albumin (g/dL) | 3.5–5.2 | 3.8 |
| Total bilirubin (mg/dL) | ≤1.23 | 0.29 |
| Alanine aminotransferase (U/litre) | ≤41 | 67 |
| Aspartate aminotransferase (U/litre) | ≤40 | 29 |
| Gamma glutamyl transferase (U/litre) | ≤59 | 194 |
| Alkaline phosphatase (U/litre) | 40–129 | 123 |
| Prothrombin time (sec) | 10.0–13.2 | 15.6 |
| International normalised ratio | 0.9–1.2 | 1.31 |

*Local laboratory reference ranges.

An extensive contact tracing exercise was undertaken in line with national guidance. This resulted in the identification of 45 medium-risk and 36 low-risk contacts, predominantly from the flight into Dublin, the travel by taxi from the airport, and the hospital ED. Classification of flight contacts was based on a modification of UKHSA guidance at the time; those seated immediately beside the case or in the three closest seats in the row in front of the case were defined as medium-risk and all others seated within three rows of the case were defined as low-risk. Medium-risk contacts were assigned to active monitoring for 21 days post-exposure and offered vaccination with modified vaccinia Ankara-Bavarian Nordic vaccine (JYNNEOS). No secondary cases were identified.

Initial communications included a national alert to clinicians and public health practitioners and a press release confirming the detection of a case of MPXV clade I in the country. Contact was made with public health authorities in Türkiye and in the DRC. An EU Early Warning and Response System alert was issued, as well as a WHO Event Information Site notification. In addition, NIMT representatives met with colleagues from the WHO Regional Office for Europe, the European Centre for Disease Prevention and Control, and the US Centres for Disease Control and Prevention, to discuss the international public health implications.

### DNA extraction and qPCR
Viral DNA was extracted using the QIAamp DNA kit (Qiagen) following the manufacturer's instructions. Sample lysis and inactivation were performed at Containment Level 3, with all subsequent steps conducted at Containment Level 2. Extracted DNA was tested for MPXV using quantitative PCR (qPCR) on the ABI ViiA 7 platform. This testing involved the concurrent use of several laboratory-developed assays—a pan-orthopoxvirus qPCR assay developed by Schroeder and Nitsche, 2010[18], clade I- and II-specific assays based on Li et al. 2010[19], and a clade Ib-specific assay performed in accordance with Schuele et al. 2024[20]—alongside a commercial qPCR assay from Altona Diagnostics (Hamburg, Germany).

## Sequencing

A skin lesion swab collected on the 5th of February 2025 with the highest titre (lowest cycle threshold (Ct) value) was selected for WGS. To maximise genome coverage and improve confidence in variant detection, sequencing was performed independently using three different MPXV primer schemes. These included the NextGenPCR MPXV amplification primer panel (MBS, the Netherlands; https://www.protocols.io/view/monkeypox-virus-whole-genome-sequencing-using-comb-n2bvj6155lk5/v1)[21], which we previously employed for the 2022-2023 MPXV clade IIb outbreak in Ireland[22], supplemented with fourteen newly-designed MPXV clade Ia-specific custom primers (Supplementary Data 1), as well as the ARTIC 2500 bp tiled scheme (https://labs.primalscheme.com/detail/artic-inrb-mpox/2500/v1.0.0/?q)[23], and the ARTIC 400 bp tiled scheme (https://labs.primalscheme.com/detail/artic-inrb-mpox/400/v1.0.0/?q)[24,25]. These tiled amplicons from the different workflows were sequenced independently on the NextSeq1000 platform with XLEAP-SBS chemistry on a P1 flow cell (Illumina) rendering 2×150 bp paired-end reads over 300 cycles, exactly as previously described[26].

## Consensus genome generation

Illumina reads were aligned to a masked clade Ia reference genome (KJ642613.1_masked, available at https://labs.primalscheme.com/detail/artic-inrb-mpox/2500/v1.0.0/?q) and processed with the MPXV Nextflow pipeline (artic-mpxv-nf, v2.0.0) developed by the ARTIC network[27]. Variant allele frequency (VAF) thresholds were adjusted so that <0.25 was called as reference, 0.25–0.75 as ambiguous (incorporated with IUPAC codes), and >0.75 as fixed variants.

## Phylogenetic analysis

Phylogenetic analysis was performed with SQUIRREL (v1.0.12)[15], using sequences downloaded from Pathoplexus (https://doi.org/10.62599/PP_SS_168.1) and GISAID, together with two outgroup sequences (KJ642617.1 and KJ642615.1). Duplicate sequences and those with >20% missing data were removed, resulting in 957 sequences for analysis. SQUIRREL's --seq-qc option identified SNVs to be masked from the alignment using the --additional-mask flag. The --run-apobec-phylo option was applied to identify mutational signatures associated with APOBEC3 activity. To obtain bootstrap estimates, IQ-TREE was used to generate a maximum likelihood tree, employing the same substitution model used by SQUIRREL (K3Pu +F + I) and performing 1000 ultrafast bootstrap replicates[28].

## Statistics and reproducibility

As a single case report, this study is descriptive in nature; therefore, no formal statistical hypothesis testing was performed. Whole-genome sequencing was performed independently three times using distinct primer schemes to maximise coverage and confidence in called variants across the runs. To enable reproducible data analysis, specific versions are provided for all software tools and analysis pipelines and the code and data used for figure generation is provided.

## Ethics

The patient provided written informed consent for publication. As this is a case report, institutional research ethics approval was not required.

## Results

### Laboratory results

On the 5th of February 2025, qPCR was performed on various sample types, resulting in the detection of both pan-orthopoxvirus DNA and MPXV clade I DNA (Table 2). Lower Ct values are indicative of higher viral loads. As expected, lower Ct values were observed in skin swabs taken directly from lesions. Higher Ct values on the 4th of February may reflect lower sample quality compared to samples from the 5th of February. Low Ct values were recorded from two skin swabs taken on the 5th of February, indicating a peak in viral replication. Over subsequent days, Ct values gradually increased or became undetectable, reflecting a declining viral load as the infection progressed. The MPXV clade Ib and MPXV clade II assays were both negative. These initial molecular diagnostic findings, together with the travel history and clinical features, indicated a probable MPXV clade Ia infection. On the 10th of February 2025, MPXV clade Ia infection was confirmed by WGS. The consensus genome (100% coverage relative to the masked reference KJ642613.1) was assigned to clade Ia using Nextclade[29], verified by confirming the presence of the MPXV *OPG032* gene, which is known to be deleted in clade Ib genomes[30].

## Phylogenetic analysis

Phylogenetic analysis revealed that the Irish sequence clustered within a group of 182 other clade Ia group II sequences (Fig. 3A, Supplementary Data 2), with strong support (bootstrap = 98%). All of these sequences were collected in 2024 and most (*n* = 171) originated in Kinshasa, DRC, with 11 from other regions of the DRC. Based on this and the presence of four mutations (G134714A, G146403A, G171616A, and G189877T; NC_003310.1 coordinates), which have been previously reported as defining mutations of the sustained human 2024 (sh2024) clade Ia outbreak (Fig. 3B), we designated this cluster as part of sh2024[14].

The Irish sequence was part of a well-supported subcluster (99% bootstrap) within sh2024, which is highlighted in green in Fig. 3. This subcluster included 20 other sequences from Kinshasa collected between the 23rd of September and the 7th of November 2024 (Supplementary Data 2).

Ten out of 61 (16%) single nucleotide variants in the Irish sequence occurred at APOBEC3-associated dinucleotide motifs (Supplementary Data 3). Of the APOBEC3-related mutations, four (G17242A, C100678T, C120444T, and G134714A) were present in sequences from the sh2024 cluster. The Irish sequence had six unique mutations, five (83%) of which were consistent with APOBEC3 editing (Fig. 3B).

## Discussion

This report describes a case of MPXV clade Ia infection detected in Dublin, Ireland, in February 2025. The patient recovered without long-term complications, and no secondary transmission was identified. To the best of our knowledge, this was the first time a case of MPXV clade Ia infection outside of Africa was confirmed and reported internationally. Two further cases of MPXV clade Ia infection have since been identified in China, in adult males with a history of travel from the DRC[31]. In addition, in July 2025, Türkiye retrospectively reported a case of MPXV clade Ia from October 2024, again in an adult male with a history of travel from the DRC[32]. These cases draw attention to issues of global significance around emergent changes in MPXV clade Ia epidemiology and transmission, the importance of genomic investigation and surveillance, and clinical aspects of this virus that are not yet fully understood.

This patient presented directly to the hospital from the airport after arriving in Ireland. On presentation, isolation and specialist review were urgently arranged based on the travel history and clinical features, and public health authorities were promptly notified. This highlights the importance of recognising risk factors and obtaining an appropriate travel history to enable timely public health actions and reduce the spread of infectious diseases. However, a recent case of MPXV clade Ib infection in the UK with no reported travel history or contact with other known cases demonstrates the importance of maintaining clinical vigilance even in the absence of clear epidemiologic links[33].

While this patient remained clinically stable throughout the illness and recovered without long-term complications, significant uncertainties exist regarding the clinical spectrum and severity of MPXV clade Ia infection. The clinical presentation of MPXV clade I commonly involves lesions on the face, extremities, and other parts of the body[13,34,35]. MPXV clade Ib infections in adults may be more commonly associated with more mucosal lesions or lesions limited to the genital area than were previously seen with clade Ia[13,36]. A recent analysis of clade Ia cases in Kinshasa also found that genital or anorectal lesions were more common than previously described[14]. The global clade IIb outbreak in 2022 was associated with frequent involvement of the anogenital area, pleomorphic lesions, and lower total lesion count[34,35,37]. Regarding clinical severity, reports suggest that MPXV clade Ia

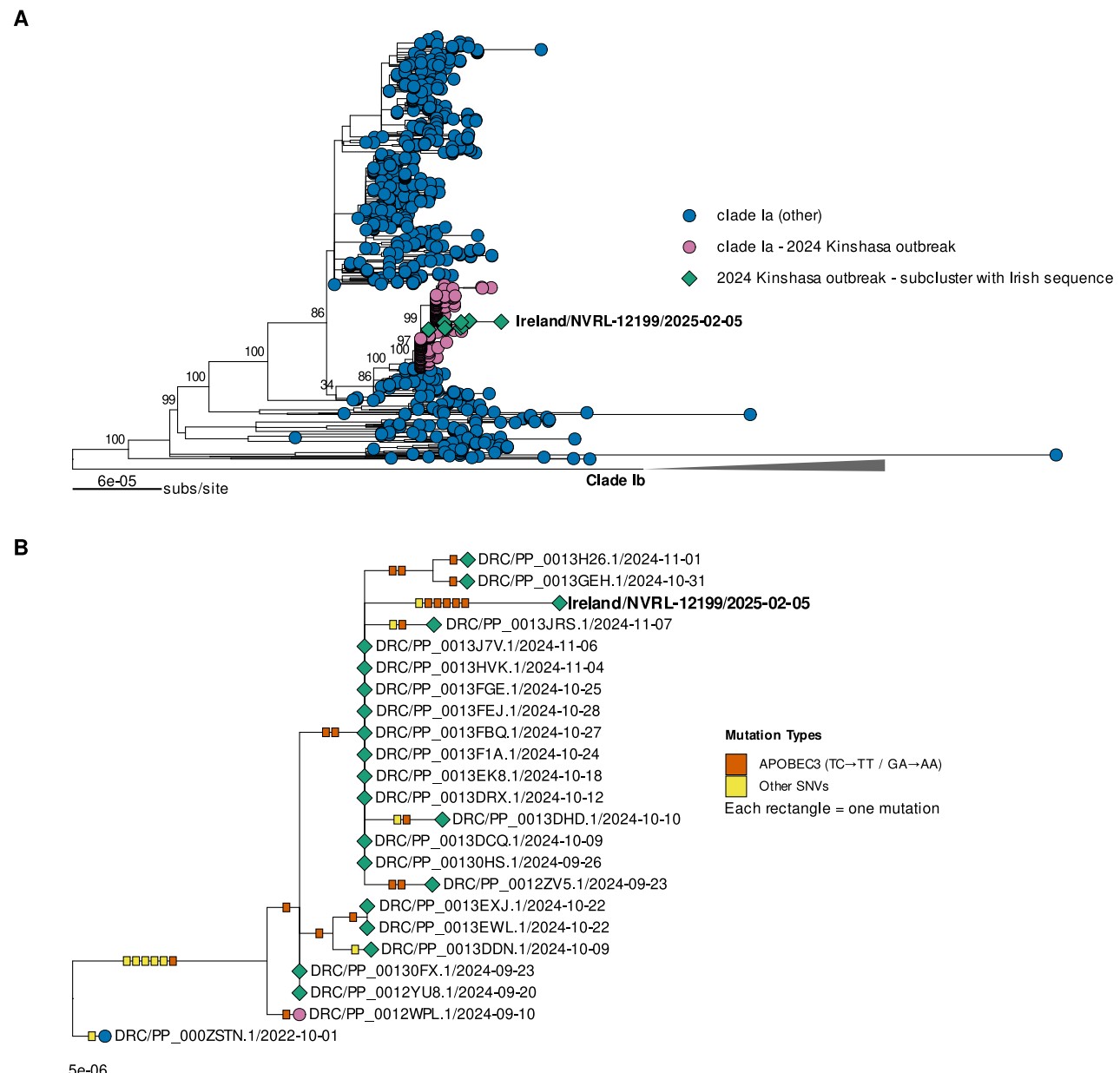

**Fig. 3 | Phylogenetic tree and APOBEC3 mutational profile of the Irish MPXV clade Ia genome imported from the Democratic Republic of the Congo. A** The maximum likelihood phylogenetic tree including the Irish sequence and clade I sequences downloaded from GISAID and PathoPlexus (n = 957). Clade Ia sequences are shown in blue, with those from the Kinshasa 2024 outbreak (sh2024) highlighted in pink. The subcluster containing the Irish sequence is marked by green diamonds. Clade Ib sequences are collapsed and represented by a grey triangle. Branch support values (bootstrap percentages) are shown at the nodes. The scale bar indicates the substitution rate per site (subs/site). **B** A zoomed-in view of the phylogenetic subcluster containing the Irish sequence, showing individual mutations along each branch. Mutations are represented as boxes, coloured by type: putative APOBEC3-induced mutations (TC → TT on the leading and GA → AA on the lagging strand) are shown in orange, and all other single-nucleotide variants (SNV) are shown in yellow. This subcluster includes 21 sequences from the sh2024 outbreak, including the Irish case. For additional context, one sequence from the broader sh2024 outbreak cluster (pink) that falls outside this subcluster, as well as one clade Ia sequence (blue), are also shown.

may be associated with a higher CFR than clade Ib or clade IIb[2]. However, the CFR in Kinshasa was recently noted to be low, at approximately 0.4%, despite the majority of sequenced cases there being clade Ia[2]. Morbidity and mortality comparisons across clades and subclades are complicated by variations in factors such as demographic profiles and the availability of medical care[8]. Certain groups may be more susceptible to more severe mpox presentations, such as people with uncontrolled HIV infection, young children, and pregnant women[13]. As such, some of the differences in observed CFRs may partly be related to differences between regions and

outbreaks, such as differences in the affected age groups[38]. Under-reporting of mild cases could also potentially contribute to a higher observed CFR in some settings. Overall, the relative contribution of clade-specific factors to the observed variations in morbidity and mortality is not yet entirely clear[39], and further clinical and epidemiological evidence is required to fully understand the morbidity and mortality associated with MPXV clade Ia.

Historically, transmission of MPXV clade Ia infection has primarily been zoonotic, with limited human-to-human spread[3]. However, recent

**Table 2 | Molecular diagnostics findings**

| Sample type | Date taken | Ct values | | | | |
|---|---|---|---|---|---|---|
| | | Pan-orthopoxvirus (laboratory-developed test) | Pan-orthopoxvirus (Altona) | MPXV clade I | MPXV clade Ib | MPXV clade II |
| Swab (Skin) | 4 Feb 2025 | 34.9 | 29.5 | 34.1 | Not Detected | Not Detected |
| Swab (Skin) | 5 Feb 2025 | 25.5 | 22.2 | 24.3 | Not Detected | Not Detected |
| Swab (Skin) | 5 Feb 2025 | 25.2 | 21.0 | 23.5 | Not Detected | Not Detected |
| Swab (Throat) | 5 Feb 2025 | 36.3 | 30.8 | 34.3 | Not Detected | Not Detected |
| Blood | 10 Feb 2025 | 43.3 | Not Tested | 39.0 | Not Tested | Not Tested |
| Swab (Skin) | 11 Feb 2025 | Not Detected | 31.4 | 38.8 | Not Tested | Not Tested |
| Swab (Throat) | 11 Feb 2025 | Not Detected | 32.6 | Not Detected | Not Tested | Not Tested |
| Urine | 13 Feb 2025 | Not Detected | Not Detected | Not Detected | Not Tested | Not Tested |

Cycle threshold (Ct) values obtained via quantitative real-time polymerase chain reaction (qPCR).

evidence indicates that the sh2024 outbreak, which is estimated to have begun in late June or early July 2024 in Kinshasa, has been driven by sustained human-to-human transmission[14]. This is supported by the observation that 68% of mutations identified in sequences from the outbreak were consistent with APOBEC3-mediated editing[14]. This Irish sequence clusters closely with these sequences. The proportion of mutations that are APOBEC3-type in this case, relative to the reference genome and the closest node in the phylogenetic tree (16% and 83%, respectively), is substantially higher than what would be expected from a zoonotic infection[3,15], despite the individual's travel history, which involved camping and close proximity to wildlife. The number of unique mutations is also higher than expected based on the estimated APOBEC3 mutation rate (~six per year) given the three–four months difference in collection dates between samples collected from this case and the most genetically similar viral genomes. This suggests that there is unsampled MPXV diversity in the human population as human-to-human transmission chains give rise to the observed APOBEC3 accumulation. However, the accumulation of APOBEC3-mediated mutations is highly stochastic and - although the observed rate has been ~six APOBEC3 mutations per year - there is variation around that, and in the absence of additional samples it is not possible to accurately estimate the true size of the outbreak[15]. Caution is therefore warranted when interpreting genomic data in the absence of additional epidemiological information; however, the sampled virus genome lies within the well-characterised sh2024 outbreak[14]. This case thus draws attention to the important epidemiological shift in MPXV clade Ia that has been recognised in the DRC and directly demonstrates the global risk of further spread through international travel.

The global outbreak caused by MPXV clade IIb predominantly involved sexual transmission between men who have sex with men[10]. Uncertainty remains as to the precise factors driving human-to-human transmission pathways in the context of the Kinshasa clade Ia outbreak, but some indirect evidence to support sexual contact as a potential mechanism includes the age distributions of cases and the involvement of the genital area in many cases, as described in a recent study[14]. Sexual transmission of MPXV clade Ia has previously been reported in a cluster of cases in the DRC[40]. Recent evidence from the DRC suggests that heterosexual transmission may be playing an important role in the spread of MPXV clade Ib[36]. Given these uncertainties and potential differences, healthcare practitioners should be aware that the epidemiological and sociodemographic characteristics of MPXV clade Ia infections may differ significantly from those associated with the clade IIb outbreak.

One limitation of this study is that cell culture and virus isolation were not available in our laboratory to supplement molecular confirmation of MPXV. However, in the absence of secondary transmission events and the relatively mild clinical course observed in this case, this was not deemed to be critical to the scope of the contact management exercise and is unlikely to have altered patient management.

## Conclusion

This report describes the key clinical and public health implications of a case of MPXV clade Ia infection detected outside of Africa. The findings draw attention to significant recent changes in MPXV clade Ia epidemiology and demonstrate the value of prompt genomic surveillance as a tool to support public health investigations and the elucidation of virus transmission routes. The case highlights the importance of heightened awareness among healthcare professionals and public health authorities in response to emerging viral threats. Ultimately, the recent and ongoing mpox outbreaks, marked by their international spread and evolving transmission pathways, underline the need for a coordinated global response.

## Data availability

The consensus genome sequence generated in this study is publicly available on Pathoplexus (Accession ID: PP_002XE2K.1) and GenBank (Accession ID: OZ261139.1). Intermediate data, including variant call information, is provided in the Supplementary Data. Raw sequencing data generated in this study are publicly available in the SRA database under project accession number PRJNA1314986. Source data underlying Fig. 3, including phylogenetic tree files, mutation annotations, squirrel output, and the R script used to generate the figure are available on Zenodo (https://zenodo.org/records/17062323)[41].

## Code availability

The R script used to generate Fig. 3 is available on Zenodo at https://zenodo.org/records/17062323[41].

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

## Acknowledgements
No funding was granted for this case report.

## Author contributions
Laboratory analysis: L.F., M.C., B.K., D.H., and C.D.G. Review of laboratory analysis: A.O.T. Writing – initial draft laboratory methods and results: L.F., M.C., B.K., and D.H. Writing – initial draft clinical description: J.H., S.M.M.F., C.K., E.M., and E.D.B. Writing – initial draft epidemiological description and public health response: J.S., A.D., E.O.D., C.R., M.O.S., A.S., S.C., M.O.M., C.D.C., D.I., and K.I.Q. Writing – initial draft introduction and discussion: C.D.C., D.I., and K.I.Q. Writing – reviewing and editing: all authors. Supervision: D.I. and D.H.

## Competing interests
The authors declare no competing interests.
