## [Transparent Peer Review file · Communications Medicine]

Mpox due to monkeypox virus clade la infection detected outside of Africa: a case study

Corresponding Author: Dr Cian Dowling-Cullen

Version 0:

Reviewer comments:

Reviewer #1

(Remarks to the Author)

The paper titled "First case of mpox due to monkeypox virus clade la infection detected outside of Africa" describes the epidemiologic clinical and laboratory features in a traveller returning to Ireland from the Democratic Republic of Congo. Also present the phylogenetic analysis and APOBEC3 – associated mutation profile. The clade la was first detected in 1970, and is endemic within countries and in central Africa, as it is the Democratic Republic of the Congo, where have been described the majority of Clade la cases. Fifty-five years later, the first case of MPXV clade la infection is detected outside the endemic African areas. This is not unexpected, because the risk of international spread by MPXV clade la was assessed moderate by WHO, however the situation in Kinshasa warrants specific focus and was linked to a higher risk of spread, at the time the patient stayed in Kinshasa.

The article addresses a relevant and current topic. Below is listed some suggestions for enriching the article, about clade la of MPXV:

Should include: a) The year when was first detected; b) The overall public health risk and international spread; c) Concerning zoonotic transmission, the animal reservoir; d) Human-to-human transmission through sexual contact; e) Estimated incubation period; f) Immunity provided by current vaccines or previous infection; g) Therapeutic considerations; h) The case fatality rate

Small corrections:

Line 67 – On the 4th February 2025...

Line 68 - ...with an one-day of a pustular rash and a six-day history of fever... (note: number from one to nine are written as words)

Line 71 - ...after a four-week visit...

Line 98 – On the 5th February 2025 (day seven of illness)...

Line 105 – By day eight of illness...

Line 107 – On day nine of illness...

Line 109 - ...after day nine of illness

Line 119 - ...on the 18th February 2025

Line 147 – On the 5th February 2025

Line 214 – (~six per year) given the three-four months...

Reviewer #2

(Remarks to the Author)

The manuscript titled "First case of mpox due to monkeypox virus clade la infection detected outside Africa" presents a timely and well-documented case report of a traveller returning to Ireland from the Democratic Republic of the Congo. The integration of clinical, epidemiological, public health, and genomic data, including APOBEC3-associated mutation analysis, is a major strength. This is a relevant and current topic, particularly as it documents, for the first time, Clade la mpox infection identified outside endemic African regions. While the case is not unexpected given the moderate risk of international spread previously assessed by WHO, the situation in Kinshasa, where the patient had travelled, warrants specific attention due to a higher potential for exportation.

These changes would strengthen the manuscript's clarity, contextual depth, and accessibility for a broad audience. The reviewers commend the authors on the value of this case and recommend revision to enhance the article's overall quality

and impact.

Below are listed some suggestions for enriching the article:

About clade Ia of MPXV the article should include in the introduction: a) The year when was first detected; b) The overall public health risk and international spread; c) Concerning zoonotic transmission, the animal reservoir; d) Human-to-human transmission through sexual contact; e) Estimated incubation period; f) Immunity provided by current vaccines or previous infection; g) Therapeutic considerations; h) The case fatality rate.

Line 58 - Consider introducing clade Ia as the main subject earlier, instead of delaying its prominence until midway through the introduction.

Line 67 – On the 4th February 2025...

Line 68 - ...with an one-day of a pustular rash and a six-day history of fever... (note: number from one to nine are written as words)Line 84 - Supplementary Table 3 is not referenced or discussed in the main text.

Line 71 - ...after a four-week visit...

Line 98 – On the 5th February 2025 (day seven of illness)...

Line 105 – By day eight of illness...

Line 107 – On day nine of illness...

Line 109 - ...after day nine of illness

Line 119 - ...on the 18th February 2025Line 135 - The absence of secondary transmission should be explicitly emphasized as a key finding in the Discussion, given its substantial relevance to public health. It should not be left implicit.

Line 147 – On the 5th February 2025

Line 169 - The APOBEC3 analysis is technically appropriate, but a brief explanation of its biological role in the introduction or in the and why it is informative for transmission dynamics would make the article more accessible.

Line 177 – It should be clarified whether or not these cases in China were linked to sexual activity.

Line 192 - It would be helpful to more explicitly compare the clinical features of clade Ia versus clade IIb or Ib infections, particularly where data are available, to better inform clinicians. While the manuscript provides a detailed description of the current case, it stops short of highlighting potential similarities or differences in clinical presentation between clades.

Line 196 - The sentence on CFR in Kinshasa (0.4%) is important but could benefit from further qualification, e.g., demographic or health system factors that may influence this figure. The authors do acknowledge in the following sentence that CFR comparisons are complicated by variations in population demographics, healthcare access, and reporting; however, these factors are not specified. For instance, a predominantly young and healthy population, low prevalence of comorbidities (such as HIV), prior smallpox vaccination coverage, or underreporting of mild cases and deaths could all contribute to a lower observed CFR. Similarly, timely access to care and adequate healthcare infrastructure may also play a role. Including specific examples would enhance the reader's understanding and contextual interpretation of the CFR figure.

Line 204 - The discussion of transmission routes is somewhat speculative but necessary. Clarify the level of evidence supporting non-sexual vs sexual transmission for clade Ia specifically.

Line 214 – (~six per year) given the three-four months...

Line 373 - Consider simplifying or better annotating Figure 3B, which, while informative, is visually dense and may be difficult for some readers to interpret.

Line 209 - The commentary on APOBEC3 mutation rates and inference of human-to-human transmission is appropriate, but could be slightly expanded to reflect its limitations (e.g., stochastic nature of mutation accumulation, possibility of unsampled intermediates). These factors may affect the accuracy of phylogenetic interpretations, and caution should be exercised when drawing definitive conclusions about transmission pathways based solely on mutational signatures

Line 388 - In Table 2, ensure that the cycle threshold (Ct) values are briefly contextualized in the Laboratory Results section, for example, what they indicate regarding viral load at different time points.

(Remarks to the Author)

The case report by Dowling-Cullen and Fahey et al. describes the first monkeypox virus clade 1a infection that was detected outside of Africa in a traveler returning from the DRC.

The report is written in good cadence and describes the case from a medical/clinical perspective first and from a microbiological/phylogenetic perspective in the latter part of the manuscript. Especially the latter part contains important observations and meets the originality requirement from a science perspective. The manuscript provides further information of the distinct transmission behavior of MPXV clade 1a viruses vs clade 1b/1b2 with apart from zoonotic- also sustained human to human transmission and without a profound risk factor predominantly for MSM as described for 1b2. Hence, the manuscript not only describes an interesting case report but also conducts full genome analysis of virus DNA isolated from the patient's lesions with subsequent phylogenetic analysis clearly showing MPXV variant clade 1a group II clustering with the Kinshasa sustained human 2024 outbreak cluster.

The report is original and of scientific merit as it adds to our understanding of MPOX virus transmission and phylogeny and I recommend it for publication.

However, I have a few minor remarks I would like to see addressed/clarified.

i) I wonder why the authors did not attempt virus isolation (Callaby et al, 2025) as it is known that there is weak evidence for diagnostic swabs' PCR cycle threshold (Ct) correlation with infectiousness (Hernandez et al, 2023; Lim et al, 2023). This could have aided with scope of contact tracing in the described public health response. I suggest that a short discussion of why this was not attempted or/and why it was maybe not necessary in this case.

ii) Further, there is no more detailed information on what basis the patient was discharged. The authors do provide the correct reference describing guidelines for discharge from the UK Health Security Agency. "De-isolation and discharge of mpox-infected patients (2025)" but from a case report point of view I feel it would have been better if the authors had included concrete information for discharge of the described case.

Version 1:

Reviewer comments:

Reviewer #1

(Remarks to the Author)

After reviewing the authors' rebuttal and revised manuscript, I acknowledge that the requested revisions were generally addressed in a thorough and appropriate manner.

I therefore recommend this manuscript for publication.

Reviewer #2

(Remarks to the Author)

I co-reviewed this manuscript with Reviewer 1

Reviewer #3

(Remarks to the Author)

The authors have addressed all of my raised remarks and concerns in a satisfying manner.

We would like to thank all of the reviewers for their consideration and constructive feedback. Please find below point-by-point responses to each of the reviewers' comments.

Reviewer #1 (Remarks to the Author):

The paper titled "First case of mpox due to monkeypox virus clade Ia infection detected outside of Africa" describes the epidemiologic, clinical and laboratory features in a traveller returning to Ireland from the Democratic Republic of Congo. Also present the phylogenetic analysis and APOBEC3 – associated mutation profile. The clade Ia was first detected in 1970, and is endemic within countries and in central Africa, as it is the Democratic Republic of the Congo, where have been described the majority of Clade Ia cases. Fifty-five years later, the first case of MPXV clade Ia infection is detected outside the endemic African areas. This is not unexpected, because the risk of international spread by MPXV clade Ia was assessed moderate by WHO, however the situation in Kinshasa warrants specific focus and was linked to a higher risk of spread, at the time the patient stayed in Kinshasa.

The article addresses a relevant and current topic. Below is listed some suggestions for enriching the article, about clade Ia of MPXV:

– Should include: a) The year when was first detected; b) The overall public health risk and international spread; c) Concerning zoonotic transmission, the animal reservoir; d) Human-to-human transmission through sexual contact; e) Estimated incubation period; f) Immunity provided by current vaccines or previous infection; g) Therapeutic considerations; h) The case fatality rate

Thank you very much for this helpful feedback. We have addressed each of these aspects in the revised version of the Introduction. Please see lines 48 – 64 and 72-79 in the revised version, copied below.

"The first human infection with clade Ia monkeypox virus (MPXV) was reported in 1970 in the Democratic Republic of the Congo (DRC).¹ Historically, most MPXV clade Ia infections have occurred through zoonotic transmission, with limited secondary human-to-human spread.^{2,3} The majority of cases have been detected in the DRC, with fewer cases reported in other central African countries.² The exact details of the mpox animal reservoir are uncertain, but research suggests that rodents are likely involved.^{4,5}

The mean incubation period of MPXV clade Ia is approximately 10 days.⁶ A review of evidence up to 2020 estimated the case fatality rate (CFR) for MPXV clade Ia at approximately 10.6%.⁷ However, more recent CFR estimates have been lower,² and these estimates may be

impacted by factors such as population demographics, healthcare access, and testing or reporting practices.⁸ Regarding specific treatments, antivirals are under evaluation, but a recent clinical trial in the DRC did not demonstrate improvement in lesion resolution with tecovirimat.⁹ The World Health Organization (WHO) clade Ia risk assessment reports cross-protection from previous orthopoxvirus infection or vaccination is likely, with moderate confidence in this assessment due to limitations in the available evidence.²

“Amid these clade IIb and clade Ib events, which have drawn considerable global attention, the epidemiology of MPXV clade Ia has also undergone significant recent changes. In particular, a shift toward increasing human-to-human transmission has been demonstrated in Kinshasa in the DRC, raising regional and international public health concerns.^{13,14} While the pathways for human-to-human transmission in this outbreak are uncertain, sexual contact may be involved.^{13,14} The WHO recently assessed the overall global public health risk associated with MPXV clade Ia as moderate, and noted that the outbreak in Kinshasa requires particular attention.¹³”

– Small corrections:

Line 67 – On the 4th February 2025...

Line 68 - ...with an one-day of a pustular rash and a six-day history of fever... (note: number from one to nine are written as words)

Line 71 - ...after a four-week visit...

Line 98 – On the 5th February 2025 (day seven of illness)...

Line 105 – By day eight of illness...

Line 107 – On day nine of illness...

Line 109 - ...after day nine of illness

Line 119 - ...on the 18th February 2025

Line 147 – On the 5th February 2025

Line 214 – (~six per year) given the three-four months...

Many thanks for noting these issues, we have corrected all of the errors noted here. These are included on lines 114, 115, 118, 145, 152, 155, 156, and 296.

Reviewer #2 (Remarks to the Author):

The manuscript titled “First case of mpox due to monkeypox virus clade Ia infection detected outside Africa” presents a timely and well-documented case report of a traveller returning to Ireland from the Democratic Republic of the Congo. The integration of clinical, epidemiological, public health, and genomic data, including APOBEC3-associated mutation analysis, is a major strength. This is a relevant and current topic, particularly as it documents, for the first time, Clade Ia mpox infection

identified outside endemic African regions. While the case is not unexpected given the moderate risk of international spread previously assessed by WHO, the situation in Kinshasa, where the patient had travelled, warrants specific attention due to a higher potential for exportation.

These changes would strengthen the manuscript's clarity, contextual depth, and accessibility for a broad audience. The reviewers commend the authors on the value of this case and recommend revision to enhance the article's overall quality and impact.

Below are listed some suggestions for enriching the article:

About clade Ia of MPXV the article should include in the introduction: a) The year when was first detected; b) The overall public health risk and international spread; c) Concerning zoonotic transmission, the animal reservoir; d) Human-to-human transmission through sexual contact; e) Estimated incubation period; f) Immunity provided by current vaccines or previous infection; g) Therapeutic considerations; h) The case fatality rate.

Thank you very much for this helpful feedback. We have covered each of these aspects in the revised version of the introduction. Please see lines 48-65 and 72-79 in the revised version, copied below.

“The first human infection with clade Ia monkeypox virus (MPXV) was reported in 1970 in the Democratic Republic of the Congo (DRC).¹ Historically, most MPXV clade Ia infections have occurred through zoonotic transmission, with limited secondary human-to-human spread.^{2,3} The majority of cases have been detected in the DRC, with fewer cases reported in other central African countries.² The exact details of the mpox animal reservoir are uncertain, but research suggests that rodents are likely involved.^{4,5}

The mean incubation period of MPXV clade Ia is approximately 10 days.⁶ A review of evidence up to 2020 estimated the case fatality rate (CFR) for MPXV clade Ia at approximately 10.6%.⁷ However, more recent CFR estimates have been lower,² and these estimates may be impacted by factors such as population demographics, healthcare access, and testing or reporting practices.⁸ Regarding specific treatments, antivirals are under evaluation, but a recent clinical trial in the DRC did not demonstrate improvement in lesion resolution with tecovirimat.⁹ The World Health Organization (WHO) clade Ia risk assessment reports cross-protection from previous orthopoxvirus infection or vaccination is likely, with moderate confidence in this assessment due to limitations in the available evidence.²”

“Amid these clade IIb and clade Ib events, which have drawn considerable global attention, the epidemiology of MPXV clade Ia has

also undergone significant recent changes. In particular, a shift toward increasing human-to-human transmission has been demonstrated in Kinshasa in the DRC, raising regional and international public health concerns.^{13,14} While the pathways for human-to-human transmission in this outbreak are uncertain, sexual contact may be involved.^{13,14} The WHO recently assessed the overall global public health risk associated with MPXV clade Ia as moderate, and noted that the outbreak in Kinshasa requires particular attention.¹³

Line 58 - Consider introducing clade Ia as the main subject earlier, instead of delaying its prominence until midway through the introduction.

Thanks for this. Clade Ia is now introduced in the first paragraph of the Introduction, on line 48.

Line 67 – On the 4th February 2025...

Line 71 - ...after a four-week visit...

Line 98 – On the 5th February 2025 (day seven of illness)...

Line 105 – By day eight of illness...

Line 107 – On day nine of illness...

Line 109 - ...after day nine of illness

Line 119 - ...on the 18th February 2025

Line 68 - ...with an one-day of a pustular rash and a six-day history of fever... (note: number from one to nine are written as words)

Line 147 – On the 5th February 2025

Line 214 – (~six per year) given the three-four months...

Many thanks for noting these. We have corrected each of the above issues. These are included on lines 114, 115, 118, 145, 152, 155, 156, and 296.

Line 84 - Supplementary Table 3 is not referenced or discussed in the main text.

We thank the Reviewer for noticing this, and have added a reference to our custom primers listed in Supplementary Table 3 to this Methods paragraph on lines 362-367 (new text added is highlighted in bold):

“The highest titre (lowest cycle threshold value) sample, from a skin lesion swab collected on the 5th February 2025, was selected for WGS on the Illumina NextSeq 1000 platform. Sequencing was performed independently three times using different MPXV primer

schemes and custom primers (Supplementary Table 3) to maximise genome coverage and improve confidence in variant detection.³³⁻³⁷ Further details are provided in the Supplementary Information.”

Line 135 - The absence of secondary transmission should be explicitly emphasized as a key finding in the Discussion, given its substantial relevance to public health. It should not be left implicit.

Thank you for this feedback. This has now been emphasised explicitly in the opening paragraph of the Discussion, on lines 237-238. This revised opening Discussion paragraph is copied below:

“This report describes the first case of MPXV clade Ia infection reported outside of the African continent. **The patient recovered without long-term complications, and no secondary transmission was identified.** Two further cases of MPXV clade Ia infection have since been identified in China, in adult males with a history of travel from the DRC.²⁰ In addition, in July 2025, Türkiye retrospectively reported a case of MPXV clade Ia from October 2024, again in an adult male with a history of travel from the DRC.²¹ These cases draw attention to issues of global significance around emergent changes in MPXV clade Ia epidemiology and transmission, the importance of genomic investigation and surveillance, and clinical aspects of this virus that are not yet fully understood.”

Line 169 - The APOBEC3 analysis is technically appropriate, but a brief explanation of its biological role in the introduction or in the and why it is informative for transmission dynamics would make the article more accessible.

We have now added the following paragraph to the Introduction to explain the role of APOBEC3 analysis in informing transmission dynamics, on lines 80-88:

“Genomic analysis has played an important role in the detection and evaluation of these emerging shifts in MPXV clade Ia epidemiology. One source of evidence supporting sustained human-to-human transmission of MPXV is the accumulation of mutational signatures consistent with apolipoprotein B mRNA editing enzyme, catalytic polypeptide-like 3 (APOBEC3) activity. These are antiviral enzymes in the human innate immune system that respond to infection by inducing specific mutations (TC→TT on the leading and GA→AA on the lagging strands) of viral genomes. The observed accumulation of APOBEC3 mutations in the viral population can thus be employed to infer the duration and extent of human-to-human transmission.¹⁵”

Line 177 – It should be clarified whether or not these cases in China were linked to sexual activity.

As far as we are aware, it has not been publicly confirmed whether or not those cases in China were linked to sexual activity.

Line 192 - It would be helpful to more explicitly compare the clinical features of clade Ia versus clade IIb or Ib infections, particularly where data are available, to better inform clinicians. While the manuscript provides a detailed description of the current case, it stops short of highlighting potential similarities or differences in clinical presentation between clades.

Thank you for this feedback. We have more explicitly described the clinical features of clade Ia, Ib, and IIb in the Discussion, on lines 257-264, as copied below.

“The clinical presentation of MPXV clade I commonly involves lesions on the face, extremities, and other parts of the body.^{13,23,24} MPXV clade Ib infections in adults may be more commonly associated with more mucosal lesions or lesions limited to the genital area than was previously seen with clade Ia.^{13,25} A recent analysis of clade Ia cases in Kinshasa also found genital or anorectal lesions were more common than previously described.¹⁴ The global clade IIb outbreak in 2022 was associated with frequent involvement of the anogenital area, pleomorphic lesions, and lower total lesion count.^{23,24,26}”

Line 196 - The sentence on CFR in Kinshasa (0.4%) is important but could benefit from further qualification, e.g., demographic or health system factors that may influence this figure. The authors do acknowledge in the following sentence that CFR comparisons are complicated by variations in population demographics, healthcare access, and reporting; however, these factors are not specified. For instance, a predominantly young and healthy population, low prevalence of comorbidities (such as HIV), prior smallpox vaccination coverage, or underreporting of mild cases and deaths could all contribute to a lower observed CFR. Similarly, timely access to care and adequate healthcare infrastructure may also play a role. Including specific examples would enhance the reader’s understanding and contextual interpretation of the CFR figure.

Many thanks for this comment. We have provided further specific examples of the factors that could influence the observed CFR, covered on lines 265-281 of the Discussion, as copied below.

“Regarding clinical severity, reports suggest that MPXV clade Ia may be associated with a higher CFR than clade Ib or clade IIb.² However, the CFR in Kinshasa was recently noted to be low, at approximately 0.4%, despite the majority of sequenced cases there being clade Ia.² Morbidity and mortality comparisons across clades and subclades are complicated by variations in factors such as demographic profiles and the availability of medical care.⁸ Certain groups may be more

susceptible to more severe mpox presentations, such as people with uncontrolled HIV infection, young children, and pregnant women.¹³ As such, some of the differences in observed CFRs may partly be related to differences between regions and outbreaks, such as differences in the affected age groups.²⁷ Under-reporting of mild cases could also potentially contribute to a higher observed CFR in some settings. Overall, the relative contribution of clade-specific factors to the observed variations in morbidity and mortality is not yet entirely clear,²⁸ and further clinical and epidemiological evidence is required to fully understand the morbidity and mortality associated with MPXV clade Ia.”

Line 204 - The discussion of transmission routes is somewhat speculative but necessary. Clarify the level of evidence supporting non-sexual vs sexual transmission for clade Ia specifically.

Thank you for this. We have further clarified the level of evidence supporting sexual transmission for clade Ia based on recent research. This is outlined on lines 312-317 of the Discussion:

“Uncertainty remains as to the exact human-to-human pathways involved in the Kinshasa clade Ia outbreak, but some indirect evidence supporting a potential role of sexual contact includes the age distributions of cases and the involvement of the genital area in many cases described in a recent study.¹⁴ Sexual transmission of MPXV clade Ia has previously been reported in a cluster of cases in the DRC.²⁹”

Line 373 - Consider simplifying or better annotating Figure 3B, which, while informative, is visually dense and may be difficult for some readers to interpret.

We thank the Reviewer for highlighting this. We have revised the legend of Figure 3B to improve clarity. This included making the heading bold, adding a definition of APOBEC3-associated mutations to the legend text, and changing "Non-APOBEC" to "Other SNVs". We also included the note "Each rectangle = one mutation" to aid interpretation.

Line 209 - The commentary on APOBEC3 mutation rates and inference of human-to-human transmission is appropriate, but could be slightly expanded to reflect its limitations (e.g., stochastic nature of mutation accumulation, possibility of unsampled intermediates). These factors may affect the accuracy of phylogenetic interpretations, and caution should be exercised when drawing definitive conclusions about transmission pathways based solely on mutational signatures

We thank the Reviewer for the feedback, and have now expanded the commentary on APOBEC3 mutations rates in the Discussion (lines 295-306) to include the additional lines below in bold:

“The number of unique mutations is also higher than expected based on the estimated APOBEC3 mutation rate (~six per year) given the three–four months difference in collection dates with the most genetically similar viral genomes, suggesting that there is unsampled diversity in the human population as human-to-human transmission chains give rise to the observed APOBEC3 accumulation. However, the accumulation of APOBEC3-mediated mutations is highly stochastic and – although the observed rate has been ~six APOBEC3 mutations per year – **there is variation around that, and in the absence of additional samples it is not possible to accurately estimate the true size of the outbreak.¹⁵ Caution is therefore warranted when interpreting genomic data in the absence of additional epidemiological data; however, the sampled virus genome lies within the well-characterised sh2024 outbreak.¹⁴**”

Line 388 - In Table 2, ensure that the cycle threshold (Ct) values are briefly contextualized in the Laboratory Results section, for example, what they indicate regarding viral load at different time points.

Thanks for recommending this. We’ve added these additional lines to our Laboratory Results section (lines 202-208):

“Lower cycle threshold (Ct) values are indicative of higher viral loads. As expected, lower Ct values were observed in skin swabs taken directly from lesions. Higher Ct values on the 4th February may reflect lower sample quality compared to samples from the 5th February. Low Ct values were recorded from two skin swabs taken on the 5th February, indicating a peak in viral replication. Over subsequent days, Ct values gradually increased or became undetectable, reflecting a declining viral load as the infection progressed.”

Reviewer #3 (Remarks to the Author):

The case report by Dowling-Cullen and Fahey et al. describes the first monkeypox virus clade 1a infection that was detected outside of Africa in a traveler returning from the DRC.

The report is written in good cadence and describes the case from a medical/clinical perspective first and from a microbiological/phylogenetic perspective in the latter part of the manuscript. Especially the latter part contains important observations and meets the originality requirement from a science perspective. The manuscript

provides further information of the distinct transmission behavior of MPXV clade Ia viruses vs clade Ib/IIb with apart from zoonotic- also sustained human to human transmission and without a profound risk factor predominantly for MSM as described for IIb. Hence, the manuscript not only describes an interesting case report but also conducts full genome analysis of virus DNA isolated from the patient's lesions with subsequent phylogenetic analysis clearly showing MPXV variant clade 1a group II clustering with the Kinshasa sustained human 2024 outbreak cluster. The report is original and of scientific merit as it adds to our understanding of MPOX virus transmission and phylogeny and I recommend it for publication.

However, I have to minor remarks I would like to see addressed/clarified.

i) I wonder why the authors did not attempt virus isolation (Callaby et al, 2025) as it is known that there is weak evidence for diagnostic swabs' PCR cycle threshold (Ct) correlation with infectiousness (Hernaes et al, 2023; Lim et al, 2023). This could have aided with scope of contact tracing in the described public health response. I suggest that a short discussion of why this was not attempted or/and why it was maybe not necessary in this case.

Virus isolation was not attempted as the laboratory does not currently perform this procedure on Hazard Group/Risk Group 3 pathogens. However, once MPXV clade Ia was identified, contact was made with the European Union Reference Laboratory for Public Health for Emerging, Rodent-borne and Zoonotic Viral Pathogens, to organise the transfer of patient samples for viral culture. This process is ongoing at the time of writing. While we believe it is unlikely that virus isolation would have significantly altered immediate decision-making in this case, we acknowledge this limitation in the context of the evidence highlighted, linking infectiousness and diagnostic swab Ct values. We thank the Reviewer for this point, and have added the following lines to our Discussion (lines 323-327):

“One limitation of this study is that cell culture and virus isolation was not available in our laboratory to supplement molecular confirmation of MPXV. However, in the absence of secondary transmission events and the relatively mild clinical course observed in this case, this was not deemed to be critical to the scope of the contact management exercise and is unlikely to have altered patient management.”

ii) Further, there is no more detailed information on what basis the patient was discharged. The authors do provide the correct reference describing guidelines for discharge from the UK Health Security Agency. "De-isolation and discharge of mpox-infected patients (2025)" but from a case report point of view I feel it would have been better if the authors had included concrete information for discharge of the described case.

Many thanks for raising this point. We agree that it would be important to include this information, and have provided a description of the specific criteria applied in the case report section, lines 165-172, as copied below:

“De-isolation and discharge planning followed Irish guidance at the time,¹⁶ which was adapted from the UK Health Security Agency (UKHSA) guidance.¹⁷ This required the patient to meet clinical criteria (clinically safe for discharge), laboratory criteria (negative PCR results from blood, urine, and throat swab), and lesion criteria (no new lesions for 48 hours, no mucous membrane lesions, all lesions to have scabbed over, the scabs dropped off, and a fresh layer of skin to have formed under all previous lesions). The patient was discharged home on day 20 of illness, on the 18th February 2025.”